# Mouse models of *Loa loa*

Nicolas P. Pionnier [1], Hanna Sjoberg [1], Valerine C. Chunda[2,3], Fanny F. Fombad[2,3], Patrick W. Chounna [2,3], Abdel J. Njouendou [2,3], Haelly M. Metuge[2,3], Bertrand L. Ndzeshang[2,3], Narcisse V. Gandjui[2,3], Desmond N. Akumtoh[2,3], Dizzle B. Tayong[2,3], Mark J. Taylor[1], Samuel Wanji[2,3] & Joseph D. Turner [1]

Elimination of the helminth disease, river blindness, remains challenging due to ivermectin treatment-associated adverse reactions in loiasis co-infected patients. Here, we address a deficit in preclinical research tools for filarial translational research by developing *Loa loa* mouse infection models. We demonstrate that adult *Loa loa* worms in subcutaneous tissues, circulating microfilariae (mf) and presence of filarial biomarkers in sera occur following experimental infections of lymphopenic mice deficient in interleukin (IL)-2/7 gamma-chain signaling. A microfilaraemic infection model is also achievable, utilizing immune-competent or -deficient mice infused with purified *Loa* mf. Ivermectin but not benzimidazole treatments induce rapid decline (>90%) in parasitaemias in microfilaraemic mice. We identify up-regulation of inflammatory markers associated with allergic type-2 immune responses and eosinophilia post-ivermectin treatment. Thus, we provide validation of murine research models to identify loiasis biomarkers, to counter-screen candidate river blindness cures and to interrogate the inflammatory etiology of loiasis ivermectin-associated adverse reactions.

[1] Centre for Drugs and Diagnostics Research, Department of Tropical Disease Biology, Liverpool School of Tropical Medicine, Pembroke Place, Liverpool L3 5QA, UK. [2] Research Foundation in Tropical Diseases and the Environment, P.O. Box 474, Buea, Cameroon. [3] Parasite and Vector Biology Research Unit, Department of Microbiology and Parasitology, Faculty of Science, University of Buea, P.O. Box 63, Buea, Cameroon. These authors contributed equally: Nicolas P. Pionnier, Hanna Sjoberg. Correspondence and requests for materials should be addressed to J.D.T. (email: joseph.turner@lstmed.ac.uk)

L oiasis (tropical eye worm) is a parasitic helminth disease affecting ~13–15 million people in forested areas of Central and West Africa[1,2]. The disease is transmitted by blood-feeding *Chrysops* tabanid flies carrying the causative agent, the filarial worm *Loa loa*. Infectious stage *L. loa* larvae develop into mature adults that migrate within subcutaneous tissues and the sub-conjunctiva. Mating adult worms release thousands of microfilarial larvae (mf) daily into the circulation, which are transmitted to the vector upon taking a blood meal[3]. Loiasis is the cause of limb oedema (Calabar swellings) following death of adult worms[4]. Chronic infections cause renal, cardiac, pulmonary and neurological pathologies[3,5,6] linked to excess mortality[7]. Loiasis is also an urgent global health problem, as severe and potentially fatal neurological serious adverse events (SAE) may occur in hypermicrofilaraemic patients (≥30,000 mf/ml blood) following annual mass drug administration (MDA) of ivermectin (IVM, Mectizan®) for the treatment of the related filarial disease, onchocerciasis (also known as river blindness)[8]. Below this hypermicrofilaraemic threshold, loiasis individuals remain at significant risk of developing non-neurological, febrile, temporarily debilitating AE following IVM treatment[8,9]. The two filarial infections overlap in Central Africa[10]. Social science investigations have identified perceived risk of loiasis AE as a major factor in persistent non-participation in onchocerciasis IVM MDA[11]. Because IVM MDA has to be delivered annually, at a coverage of 80%, for periods of 15 years or more, to prevent onchocerciasis transmission[12], and because in loiasis-endemic Central African foci, elimination is not estimated to occur deploying this strategy until >2045[13], co-infection poses a significant barrier for onchocerciasis elimination programmes. There is a pressing need to develop new treatment strategies for both loiasis and onchocerciasis to de-risk AE occurrence in loiasis co-endemic areas and increase participation in river blindness elimination campaigns. Strategies include developing a safe drug cure, which selectively targets adult *Onchocerca* and/or *Loa* without inducing the rapid *Loa* microfilaricidal activity of IVM[14] and/or accurately diagnosing loiasis hypermicrofilaraemic individuals at risk of adverse reactions in a test-and-not-treat with IVM strategy[15].

*L. loa* naturally infects an endangered species of monkey, the drill (*Mandrillus leucophaeus*), endemic to Central Africa. As a surrogate non-human primate model, the life cycle of *L. loa* can be maintained via experimental infection of splenectomised baboons[16,17], a model which has recently been utilised to initiate exploratory pathology studies of IVM-associated neurological SAE[18,19]. This model has also generated adult and mf parasitic stages for a range of ex vivo studies. However, throughput of the baboon model is severely constraining for anti-filarial drug research and to identify potential targets for adjunct therapies to prevent IVM SAE. There is currently no microfilaraemic small animal model of loiasis to use as a refinement to non-human primates for loiasis translational research whilst simultaneously increasing throughput of preclinical candidate evaluations. Recently, we identified that attenuated development of juvenile adult *L. loa* from infectious inoculates could be generated in selective immune knockout mouse strains with impaired interleukin-4, IL-5 and IL-13 signalling, providing proof-of-concept that targeting host-adaptive immunity could allow development of human *Loa* isolates in mice[20]. Further, we have recently established immunodeficient mouse models of related filarial infections (*Brugia* and *Onchocerca*) and implemented them as preclinical macrofilaricidal drug screens[21–25].

Here, we address the current limitation in loiasis preclinical infection models by evaluating whether lymphopenic, immunodeficient mouse strains are susceptible to patent infection with *L. loa*. Further, we explore whether stage-specific, *L. loa* microfilaraemic mice can be established by blood infusion as a more rapid and facile model system for drug screening, including use of immunocompetent mice for the purposes of exploring inflammatory responses post-IVM treatment.

## Results

**Lymphopenic γc-deficient mice are permissive hosts of *L. loa*.** As with other human filarial parasites, fully permissive *L. loa* infections cannot establish in immunocompetent mice due to type-2 associated immunity[21,26]. Developing larvae can survive up to 2 months post infection in IL-4Rα$^{-/-}$/IL-5$^{-/-}$ BALB/c mice, but sexual maturity and production of microfilariae (mf) is not evident in these selective type-2 cytokine deficient animals[20]. We have defined the minimum pre-patent period prior to mf release into blood as 5 months post infection with human strain *L. loa* larvae in baboons[18]. We therefore investigated the long-term parasitological success of *L. loa* infection in a panel of 'severe-combined' lymphopenic immunodeficient mice (Fig. 1). Moderate levels of pre-patent adult *L. loa* infection were evident in CB.17 (BALB/c congenic) SCID mice (8/9 mice infected, median % recovery of inoculate = 8.5, Fig. 1a) 3 months post infection. Most NOD.SCID mice had cleared infection at the same time point (2/6 mice infected), suggesting a background strain-dependent susceptibility in CB.17(BALB/c) versus NOD lymphopenic mice (Fig. 1a). However, deletion of the common IL-2/7 gamma chain (γc) on the NOD.SCID background rendered mice highly susceptible to infection at the pre-patent adult stage (73% median recovery, 4/4 mice infected, $p < 0.01$ when comparing NOD.SCIDγc$^{-/-}$ and CB.17 SCID mice, Kruskal–Wallis with Dunn's post-hoc test, Fig. 1a). We therefore infected both NOD.SCIDγc$^{-/-}$ mice (commonly known as the NOD.SCID Gamma or NSG research model) and another compound immunodeficient lymphopenic mouse line on the BALB/c background, RAG2$^{-/-}$γc$^{-/-}$, and compared parasitological success at the time point of expected patency (5 months) with RAG2$^{-/-}$ γc-sufficient mice. Both compound gamma chain-deficient mouse lines supported survival of adult *L. loa* at +5 months (NOD.SCIDγc$^{-/-}$ 4/4 mice infected, 25.5% median recovery, BALB/c RAG2$^{-/-}$γc$^{-/-}$ 9/9 mice infected, 13% median recovery). Conversely, all BALB/c RAG2$^{-/-}$ mice had cleared infection at 5 months (0/10 mice infected, $p < 0.0001$ when comparing BALB/c RAG2$^{-/-}$γc$^{-/-}$ to BALB/c RAG2$^{-/-}$ mice, Mann–Whitney test, Fig. 1a). At this time point in both compound immunodeficient mouse strains, the majority of worms were found in the natural tissue niches of *Loa* adult stages, namely the subcutaneous and muscle fascia tissues (88 and 78% of total recovered worms in NOD.SCIDγc$^{-/-}$ and BALB/c RAG2$^{-/-}$γc$^{-/-}$ mice, respectively, $n = 4$–11, Fig. 1b). Mature male and female worms were recovered, determined by marked difference in lengths (2.9 ± 0.3 cm males, vs. 4.5 ± 0.3 cm females) combined with distinct reproductive morphological characteristics, at a ratio of between 3:1 and 4:1 females per male ($n = 4$–11, Fig. 1c, d; Supplementary Fig. 1, Movie 1). Development of patency was apparent via identification of all embryonic stages within female uteri ($n = 4$–11, Fig. 1d) and release of motile mf ex vivo (Supplementary Movie 2). Further, microfilaraemias in BALB/c RAG$^{-/-}$γc$^{-/-}$ mice 5 months post infection were apparent (Supplementary Fig. 1).

Due to the high reproducible parasitological success of mature *L. loa* in compound deficient mice at 5 months post infection, we surgically implanted recovered female and male worms ($n = 5$ per sex) subcutaneously into NOD.SCIDγc$^{-/-}$ or BALB/c RAG2$^{-/-}$ recipient mice. Adult motility under the skin of recipient mice was frequently evident (Supplementary Movie 3). One month post *L. loa* adult implantation (6 months

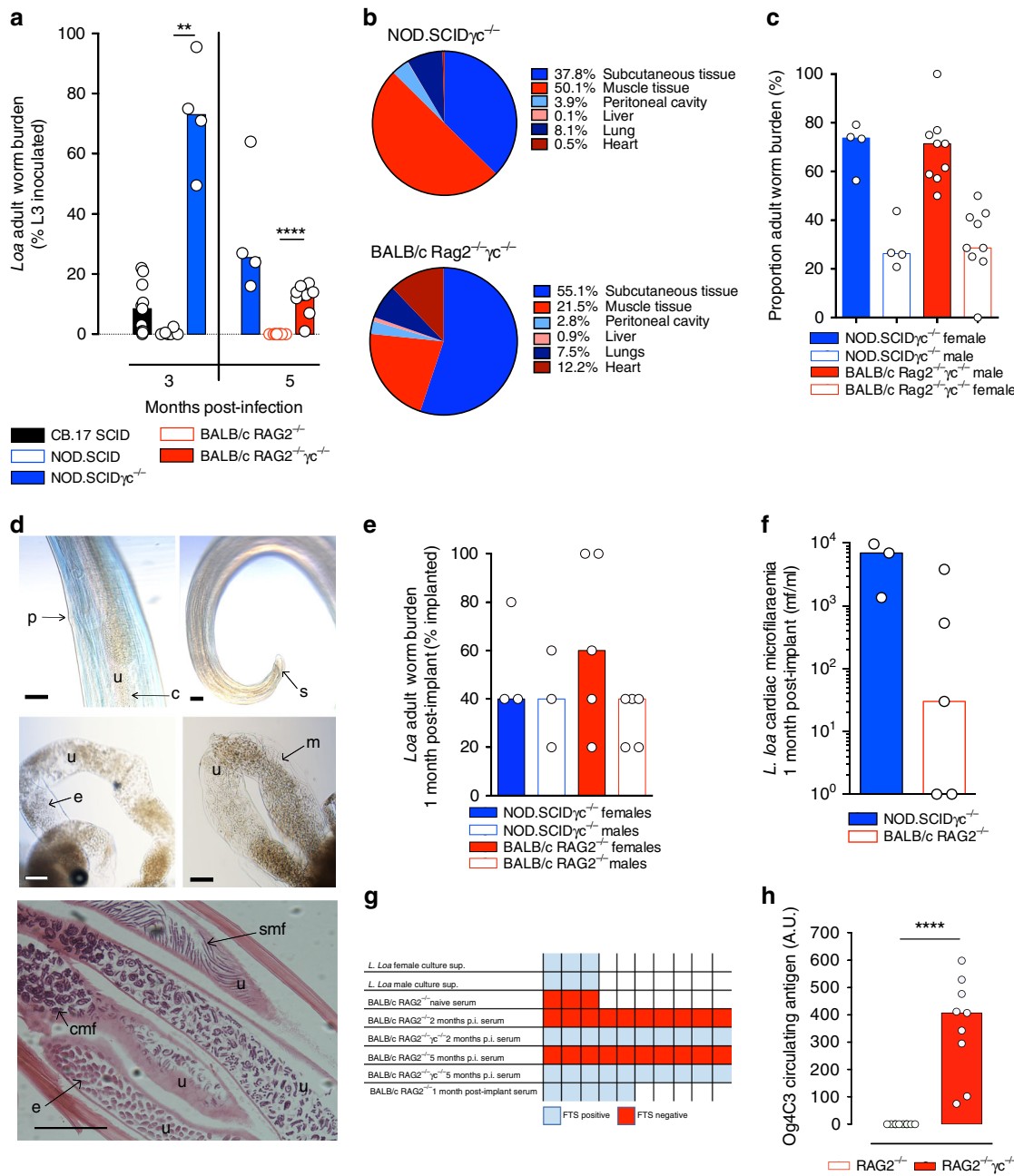

**Fig. 1** Chronic *L. loa* infection can be achieved in lymphopenic mice deficient in the common gamma chain. **a** *L. loa* adult worm burdens at 3 and 5 months post infection in lymphopenic mouse strains with or without compound IL-2/7 gamma chain (γc) deficiency (3 months: CB.17 SCID ($n = 11$), NOD.SCID ($n = 6$), NOD.SCIDγc$^{-/-}$ ($n = 4$). 5 months: NOD.SCIDγc$^{-/-}$ ($n = 4$), BALB/c RAG2$^{-/}$ ($n = 10$), BALB/c RAG2$^{-/-}$γc$^{-/-}$ ($n = 9$ mice)). Plotted is percentage adult recovery of infectious inoculum for individual mice and median levels. **b** Tissue distributions of adult *L. loa* in NOD.SCIDγc$^{-/-}$ ($n = 4$) or BALB/c RAG2$^{-/-}$γc$^{-/-}$ ($n = 9$) mice, 5 months post infection. Plotted is mean percentage of total yields per tissue site. **c** Sex ratio of adult *L. loa* in NOD.SCIDγc$^{-/-}$ ($n = 4$) or BALB/c RAG2$^{-/-}$γc$^{-/-}$ ($n = 9$) mice, 5 months post infection. Plotted is percentage adult recovery of infectious inoculum for individual mice and median levels. **d** Representative photomicrographs of *L. loa* female (upper left and middle pictures) and male (upper right) worms recovered from an infected BALB/c Rag2$^{-/-}$γc$^{-/-}$ mouse at 5 months post infection. u uterus, p uterine pore, c coiled microfilariae, m microfilariae, s spicule. Scale bar = 1 cm. The micrograph at the bottom represents an hematoxylin/eosin staining of a paraffin-embedded section of a *L. loa* female worm recovered 1 month post implant in a BALB/c Rag2$^{-/-}$ mouse. u uterus, e embryos, smf stretched microfilariae, cmf coiled microfilariae. Scale bar: 0.2 mm. **e** Adult worm yield 1 month post adult implants in NOD.SCIDγc$^{-/-}$ and BALB/c Rag2$^{-/-}$ mice, $n = 3$–5. **f** Microfilariae production 1 month post-adult implants in NOD.SCIDγc$^{-/-}$ ($n = 3$) or BALB/c Rag2$^{-/-}$ mice ($n = 5$). **g** Summary table of filariasis test strips (FTS) outcome when testing worms culture supernatants after overnight incubation at 37 °C ($n = 3$ cultures) and mice sera from selected infection conditions, ($n = 3$ naive mice, $n = 5$ BALB/c Rag2$^{-/-}$ mice 1 month post implantation, $n = 10$ all other mouse infections). **h** Og4c3 quantitative circulating antigen at 5 months post infection, ($n = 10$ BALB/c Rag2$^{-/-}$ or $n = 9$ BALB/c RAG2$^{-/-}$γc$^{-/-}$ mice). Plotted is individual data and median levels. Significant differences between strains are determined by two-tailed Mann–Whitney (two groups) or Kruskal–Wallis with Dunn's post-hoc tests (>two variables). Significance is indicated: *$p < 0.05$, **$p < 0.01$, ****$p < 0.0001$

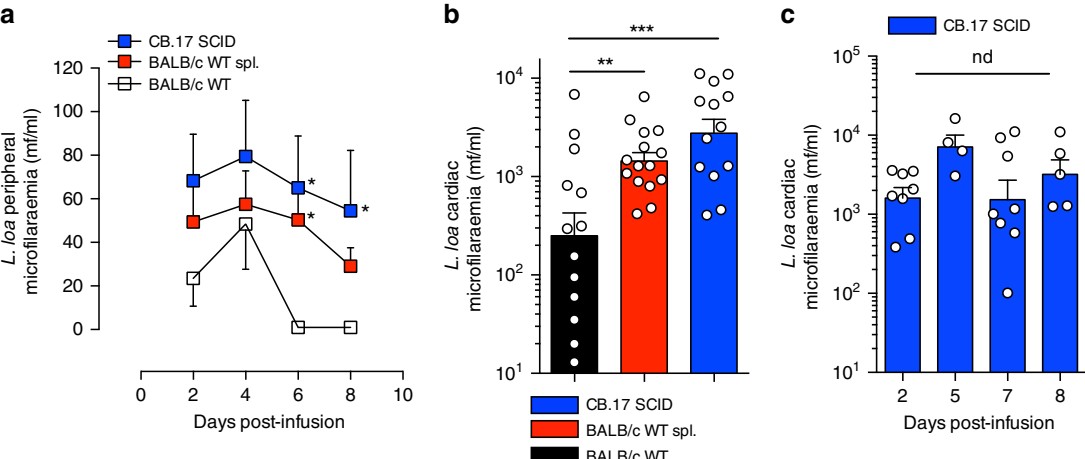

**Fig. 2** *L. loa* microfilariaemias can be established in immunocompetent and immunocompromised mice. **a** Time-course of peripheral microfilariaemias in CB.17 SCID ($n = 12$), BALB/c WT adult splenectomised (spl.), $n = 14$ or BALB/c WT, $n = 8$ mice, 2–8 days following infusion with 40,000 purified *L. loa* mf. Plotted are mean ± SEM of time course data. **b** Level of *L. loa* microfilariaemia in cardiac blood in CB.17 SCID, $n = 13$, BALB/c WT spl., $n = 13$ or BALB/c WT, $n = 15$ mice 7 days post infusion with 40,000 purified *L. loa* mf. Plotted are individual data and means ± SEM. **c** Level of *L. loa* microfilariaemia in cardiac blood in groups of $n = 4$–8 CB.17 SCID microfilaraemic mice, evaluated at between 2 and 8 days following infusion with 40,000 purified *L. loa* mf. Plotted are individual data and means ± SEM. All data are derived from a single individual experiment (**c**) or pooled from two experiments (**a**, **b**). Significant differences are determined by one-way ANOVA with Dunnett's post-hoc tests for >2 groups of log-transformed data (**a**, **b**) or Kruskal–Wallis with Dunn's post-hoc tests, per time point (**c**). Significance is indicated: *$p < 0.05$, **$p < 0.01$, ***$p < 0.0001$, nd not different

development from infectious stage larvae), viable males and females could be recovered, mainly from subcutaneous tissues, in both mouse strains (3/3 NOD.SCID $\gamma c^{-/-}$ and 5/5 BALB/c RAG2$^{-/-}$ recipients, Fig. 1e; Supplementary Fig. 2 and Movie 4). Furthermore, females were reproductively active 1 month following implantation assessed by embryograms ex vivo (Supplementary Fig. 2) with 3/3 NOD.SCID $\gamma c^{-/-}$ (mean 5980 ± 2240 mf/ml) and 3/5 BALB/c RAG2$^{-/-}$ (mean 874 ± 740 mf/ml) recipients displaying microfilariaemias by this stage (Fig. 1f).

We tested the utility of the loiasis mouse model to detect biomarkers of living *L. loa* adult infection by using two commercially available kits originally developed to specifically detect *Wuchereria bancrofti* circulating filarial antigens: the Alera Filarial Test Strip (FTS) and the TropBio Og4C3 immunoassay. Corroborating reported cross-reactivity of the FTS in recognising *Loa* specific secreted antigens[27,28], we identified a strong positive signal in supernatants of both female and male *Loa* ex vivo cultures (Fig. 1g; Supplementary Fig. 3). When examining sera from mice infected with *L. loa* L3, we found that antigenaemia detection with FTS was reproducibly apparent, dependent on *Loa* adult infection status, but independent of the age of adult worm infection or microfilaraemic status. Negative serology results were recorded in all naïve BALB/c RAG2$^{-/-}$ mice (0/3 mice tested) or in BALB/c RAG2$^{-/-}$ mice that had cleared *L. loa* at 2–5 months post infection (0/10 mice FTS positive at each time point). However, all BALB/c RAG2$^{-/-}$$\gamma c^{-/-}$ infected mice (10/10 mice, 2 or 5 months post infection) and all BALB/c RAG2$^{-/-}$ implanted with adult *L. loa* (5/5 mice, one month post implantation) were FTS positive (Fig. 1g; Supplementary Fig. 3). Additionally, all infection positive BALB/c RAG2$^{-/-}$$\gamma c^{-/-}$ mice (8/8 mice) but not infection negative RAG2$^{-/-}$ mice (0/9 mice) were seropositive for the distinct filarial adult antigen, Og4C3, 5 months after experimental infection ($p < 0.0001$, Mann–Whitney test, Fig. 1h).

**Loa parasitaemias can establish in mice.** Due to the complexity and long lead-time prior to emergence of microfilaraemias in experimental infections/implantations of immunodeficient mice,

we assessed whether infusion with purified *L. loa* mf could establish stable parasitaemias as a more facile, stage-specific infection model. We also compared performance of immunocompetent versus immunodeficient mouse lines. After administering intravenous injections of $4 \times 10^4$ *L. loa* mf through the tail vein, we observed that wild-type (WT) BALB/c mice had scant and transient microfilaraemias in peripheral blood up to 4 days post infusion (mean 48 ± 21 mf/ml, $n = 8$) after which time, no peripheral circulating mf were detectable (0/8 mice, Fig. 2a). Conversely, *L. loa* microfilaraemias were consistently identified in cardiac blood samples at termination, at 7 days post infusion, and at higher densities compared with peripheral parasitaemias (mean 1073 ± 503 mf/ml, $n = 13$, Fig. 2b). This indicated that *L. loa* mf sequestered in the cardiopulmonary circulation, with only a minor transient peripheral circulation. To test whether splenic secondary lymphoid tissue or global adaptive immune status exerted a role in limiting *L. loa* microfilaraemias in vivo, we compared parasitaemias in WT adult splenectomised or SCID mice post infusion with $4 \times 10^4$ *L. loa* mf. Splenectomy of BALB/c mice allowed a significantly ~twofold higher cardiac microfilaraemia at 7 dpi versus WT mice (mean 1894 ± 411 mf/mL, $p = 0.01$, one-way ANOVA with Dunnett's post-hoc test, $n = 13$, Fig. 2b) and *L. loa* mf persisted in peripheral blood (assessed until 8 dpi, Fig. 2a). In CB.17 SCID mice, peripheral parasitaemias were also consistently detectable (Fig. 2a, $n = 8$). SCID *L. loa* cardiac microfilaraemias were further elevated ~fourfold versus WT mice (mean 4562 ± 1098, $p = 0.001$, one-way ANOVA with Dunnet's post-hoc test, $n = 15$, Fig. 2b), with cardiac parasitaemias consistent and stable over days 2–8 in a time course experiment ($n = 4$–8 per time point, Fig. 2c). Splenectomy of SCID mice did not further increase yields of microfilaraemias compared with SCID controls (Supplementary Fig. 4).

**Loa microfilaraemic mice respond to ivermectin.** We have recently developed and implemented CB.17 SCID mouse models of lymphatic filariasis and onchocerciasis to evaluate candidate macrofilaricidal activities[21–25]. We have also demonstrated a rapid microfilaricidal response of IVM against circulating *Brugia malayi* mf in this research model[21,24]. Considering the advantage

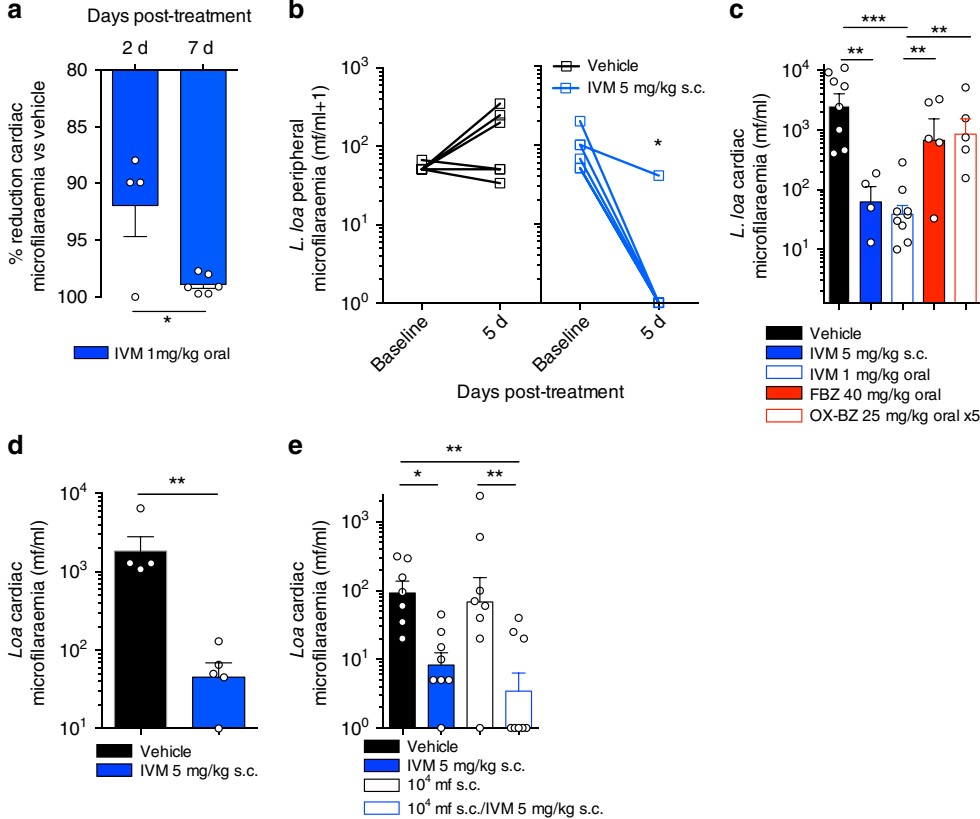

**Fig. 3** Ivermectin mediates rapid microfilaricidal activity in *L. loa* microfilaraemic mice. **a** Level of microfilaricidal efficacy in CB.17 SCID mice, 2 days ($n = 4$) or 7 days ($n = 6$) post-single oral treatment with ivermectin (IVM), expressed as a percentage reduction in cardiac blood *L. loa* microfilariaemias from mean vehicle control levels. Plotted are individual data and means ± SEM. **b** Change in peripheral *L. loa* in CB.17 SCID microfilaraemic mice ($n = 6$), 5 days following vehicle or single IVM injection. Plotted are individual data. **c** Differential microfilaricidal efficacy in microfilaraemic CB.17 SCID mice of oral ($n = 8$) and parenteral (subcutaneous; s.c, $n = 4$) IVM dosing compared with macrofilaricidal benzimidazole drugs, flubendazole (FBZ, single dose, $n = 5$) or oxfendazole (OX-BZ, daily for 5 days, $n = 5$). Plotted are individual data and means ± SEM. **d** Microfilaricidal efficacy of parenteral IVM dosing ($n = 5$) versus vehicle ($n = 4$) in microfilaraemic splenectomised BALB/c mice. Plotted are individual data and means ± SEM. **e** Microfilaricidal efficacy of parenteral IVM dosing in microfilaraemic BALB/c mice ($n = 8$ per group) ± prior immune priming (2 weeks before treatment) with s.c. inoculations of $10^4$ heat-killed *L. loa* mf or in vehicle controls ($n = 7$). Plotted are individual data and means ± SEM. Data are pooled or representative of two individual experiments. Significant differences are determined by Student's *T* test for two groups or one-way ANOVA with Dunnett's post-hoc tests for >2 groups of log10 transformed data. Significant changes in paired data are tested by Wilcoxon Tests. Significance is indicated: *$p < 0.05$, **$p < 0.01$, ***$p < 0.0001$

of a unified 'pan-filarial' research model to interrogate PK/PD relationships of candidate macrofilaricidal drugs, including off-target effects on *Loa* mf, and because the highest titres of *Loa* mf could be achieved in CB.17 SCID mice, we selected this lymphopenic mouse strain for validation assessments as a loiasis microfilaraemic counter-screening model. To this end, CB.17 SCID mice were infused with $4 \times 10^4$ *L. loa* mf and treated with single-dose IVM or vehicle. IVM mediated a rapid clearance of mf at 2 days post treatment (92% mean cardiac load reduction versus vehicle, $n = 4$, Fig. 3a). This significantly increased to a 99% mean reduction versus vehicle at 7 days post treatment (Student's *T* test, $p = 0.013$, $n = 6$, Fig. 3a). Peripheral microfilaraemias were completely cleared in 5/6 mice following treatment at 5 dpi (Wilcoxon test, $p = 0.031$, Fig. 3b). These rapid IVM response dynamics mirrored the efficacy typically observed in both experimentally infected baboons and human patients[3,19]. The rapid in vivo microfilaricidal activity of IVM was not emulated following short-course oral treatments with either of the candidate macrofilaricide drugs, flubendazole or oxfendazole[24,29], verifying a lack of off-target efficacy of these benzimidazole chemotypes against *Loa* microfilaraemias (Fig. 3c). Contrastingly, IVM efficacy against *Loa* mf was not emulated in vitro unless ≥ 2-day exposures exceeded, by 1000-fold, the

typical peak plasma concentrations in vivo following oral dosing (set at 40 ng/ml, Supplementary Fig. 5). Similarly, the more potent macrocyclic lactone, moxidectin, failed to mediate any direct toxicity in vitro against *Loa* mf at physiologically relevant levels (Supplementary Fig. 5). The in vitro insensitivity of *Loa* mf to macrocyclic lactones was similar to that of the human filaria, *Brugia malayi* (Supplementary Fig. 5).

Due to the stable parasitaemias evident in both splenectomised and non-splenectomised, immune competent BALB/c WT mice, we investigated whether IVM microfilaricidal efficacy was potentiated by host adaptive immune status. When microfilariaemic splenectomised BALB/c mice were treated with IVM, a similar degree of efficacy was observed compared with SCID mice (mean 98% reduced microfilaraemia, 7 days post treatment, $p = 0.011$, Student's *T* test, $n = 4–5$, Fig. 3d). Immune-priming BALB/c mice with subcutaneous inoculations of heat-killed *L. loa* mf 2-weeks prior to infusion and IVM treatment augmented the IVM treatment response (90% vs. 97% mean reduction in naive versus immune-challenged mice 7 days post treatment, $n = 8$ per group, one-way ANOVA followed by Dunnett's post-hoc test, $p < 0.05$ versus $p < 0.01$ compared with respective vehicle controls, Fig. 3e). Together the data indicate that *L. loa* microfilaraemic mice display a typical rapid initial microfilaricidal drug response

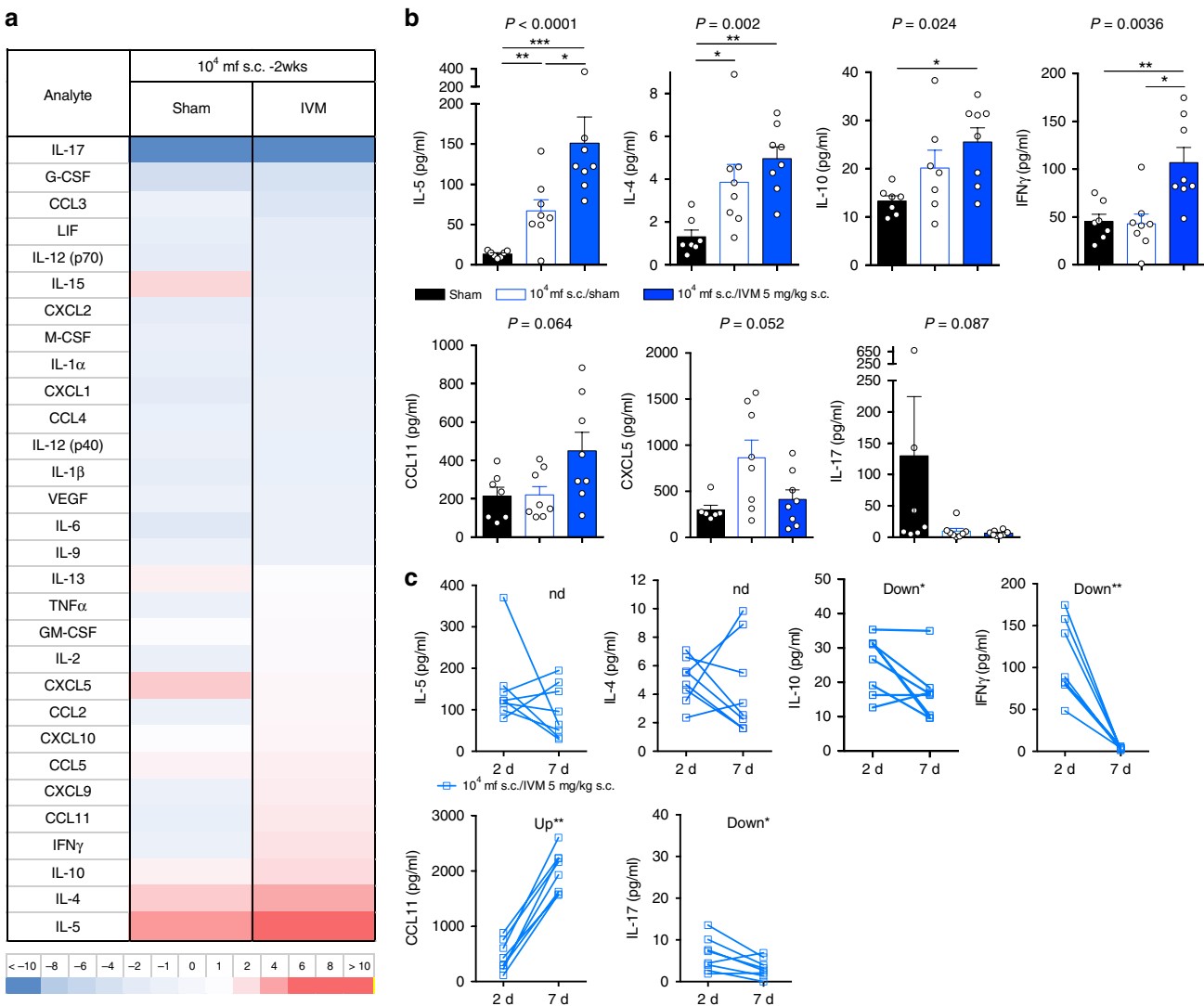

**Fig. 4** Type-2 systemic inflammation is evident post-ivermectin treatment of *L. loa* microfilaraemic mice. **a** Plasma cytokines/chemokine changes 2 days post infusion with 40,000 purified *L. loa* mf into BALB/c mice with prior immune-priming via s.c. inoculations of $1 \times 10^4$ heat-killed *L. loa* mf and either IVM 5 mg/kg ($n = 8$) or sham-treatment s.c. at the point of infusion ($n = 8$). Plotted is median fold change compared with non-immune primed, sham-treated *L. loa* microfilaraemic controls ($n = 7$). Colour scale depicts $< -10$ (dark blue) to $>10$ (dark red) fold difference with white being 1 (i.e., no difference). **b** Cytokines/chemokines with ≥twofold increase or decrease following immune-priming ($n = 8$) and/or immune-priming + IVM treatments ($n = 8$) compared with non-immune primed, sham-treated *L. loa* microfilaraemic controls ($n = 7$). Data plotted are individual data and means ± SEM. Significant differences are determined by one-way ANOVA with Dunnett's post-hoc tests of either raw or log-transformed data (IL-4, 5, 10, CCL11, CXCL5) or Kruskal–Wallis tests with Dunn's post-hoc tests (IFNg, IL-17). **c** Changes in circulating cytokine/chemokine levels between 2 and 7 days post-IVM treatment in immune-primed *L. loa* microfilaraemic mice ($n = 8$). Plotted are individual data. Significant differences are determined by paired Student's T tests (IL-4, -5, -10) or Wilcoxon tests (IFNg, CCL11, CXCL5, IL-17). All data are pooled from two individual experiments. Significance is indicated: *$p < 0.05$, **$p < 0.01$. ***$p < 0.001$

to IVM over the course of 7 days post treatment and efficacy remains intact in SCID or asplenic mice. However, we provide evidence that prior exposure to *Loa* mf antigens in immune competent mice can bolster the rapid efficacy of IVM.

**Ivermectin treatment of *L. loa* induces type-2 inflammation.** Taking advantage of stable parasitaemias over 7 days in WT mice following immune-priming with heat-killed *L. loa* mf, we undertook luminex bead-based array analysis of 32 serological cytokines/chemokines 2–7 days post-IVM treatment and compared responses to untreated immune-primed mice or immunologically naive mice that had received matching infusions of $4 \times 10^4$ *L. loa* mf ($n = 8$ mice per group, Fig. 4). Analytes that were up or downregulated by more than twofold in immune-

primed mouse groups ( + / − IVM treatment) versus naive mice 2 days following *L. loa* infusions were interleukins (IL)-4, 5, 10, 17, interferon gamma (IFNγ), CC-chemokine ligand 11 (CCL11; eotaxin 1) and CXC-chemokine ligand 5 (CXCL5, Fig. 4a). These analytes were then examined in statistical models to determine significance of fold-changes following immune-priming and IVM treatment. Whilst immune-priming alone induced significant elevations in circulating interleukins (IL)-4 and IL-5 2 days following mf infusions compared with control mice (mean 3.9 ± 0.8 vs. 1.3 ± 0.3 pg/ml, $p < 0.05$ and 67 ± 13.9 pg/ml vs. 13.5 ± 1.5, $p < 0.01$), the magnitude of the IL-4 and IL-5 response was bolstered following IVM treatment (mean 5.0 ± 0.5 pg/ml, $p < 0.01$ and 151.2 ± 32.5 pg/ml, $p < 0.001$, one-way ANOVA followed by Dunnett's post-hoc tests, Fig. 4b). Levels of IL-10 and IFNγ were

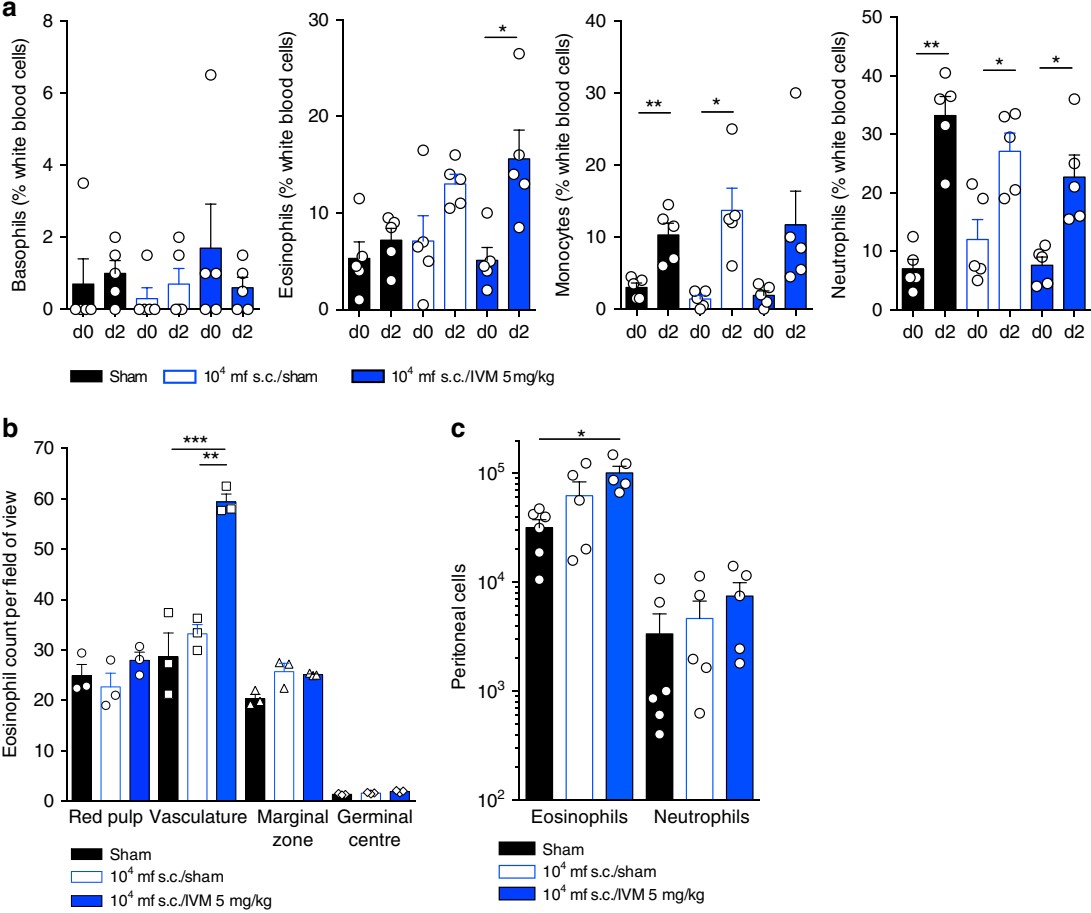

**Fig. 5** Ivermectin treatment induces eosinophilia in *L. loa* microfilaraemic mice. **a** Differential white blood cell counts determined by blood thin smear and MGG staining at baseline or 2 days post infusion with 40,000 purified *L. loa* mf into BALB/c mice ± prior immune-priming with s.c. inoculations of $1 \times 10^4$ heat-killed *L. loa* mf and sham or IVM 5 mg/kg treatment s.c. at the point of infusion (*n* = 5 mice per group). Proportions of basophils, eosinophils, monocytes and neutrophils were enumerated per 200 leucocytes per slide. Data plotted are specific cell proportions of total leucocytes (%) of individual mice and mean ± SEM. **b** Eosinophil counts in the red pulp, vasculature, white pulp marginal zones and germinal centres in sections of spleen 7 days post infusion with 40,000 purified *L. loa* mf into BALB/c mice ± prior immune-priming with s.c. inoculations of $1 \times 10^4$ heat-killed *L. loa* mf and sham or IVM 5 mg/kg treatment s.c. at the point of infusion (*n* = 3 mice per group). Plotted is mean counts for 10 quadrants of 120 μm x 200 μm per splenic zone per mouse, and mean ± SEM per group. **c** Eosinophil and neutrophil enumerations within peritoneal exudate cells acquired by flow cytometry, 7 days post infusion with 40,000 purified *L. loa* mf into BALB/c mice ± prior immune-priming with s.c. inoculations of $1 \times 10^4$ heat-killed *L. loa* mf and sham or IVM 5 mg/kg treatment s.c. at the point of infusion (*n* = 5 mice per group except sham group, *n* = 6). Plotted is individual data and mean ± SEM. Significant changes in paired data is tested by *T* test (**a**). Significant differences for three independent groups are determined by one-way ANOVA with Dunnett's post-hoc tests (**b**, **c**). Significance is indicated: *$p < 0.05$, **$p < 0.01$, ***$p < 0.001$

only significantly elevated in the circulation of immune-primed plus IVM-treated microfilaraemic mice compared with controls (mean 25.5 ± 2.3 vs. 13.3 ± 1.1 pg/ml, *p* < 0.05 and 107.0 ± 15.9 vs. 45.4 ± 7.5 pg/ml, *p* < 0.01, one-way ANOVA followed by Dunnett's post-hoc tests or Kruskal–Wallis test followed by Dunn's post-hoc tests, respectively, Fig. 4b). Due to augmented efficacy of IVM after 7 days in immune-primed mice (Fig. 3), we examined the change in inflammatory responses that were initially up or downregulated post-IVM treatment between D2 and D7 post treatment. Levels of circulating IL-4 and IL-5 were not altered by D7, whereas IL-10 (mean 16.7 ± 2.9 vs. 25.5 ± 2.3 pg/ml, *p* = 0.036, Wilcoxon test), IFNγ (mean 3.7 ± 0.5 versus 107.0 ± 15.9 pg/ml, *p* = 0.008, Wilcoxon test) and IL-17 (mean 3.3 ± 0.8 vs. 6.5 ± 1.4 pg/ml, *p* = 0.039, Wilcoxon test) all significantly had decreased by this time point post-IVM treatment (Fig. 4c). Contrastingly, circulating levels of CCL11 had increased on average by > fourfold between D2 and D7 post-IVM treatment (1993 ± 134.4 vs. 450.8 ± 96.5 pg/ml, *p* = 0.008, Wilcoxon test). Therefore, a type-2 'eosinophilic-like' inflammatory signature was

evident in immunocompetent, antigen-experienced animals 7 days following treatment with IVM. Differential white blood cell counts were compared at baseline and D2 in individual mice. Both monocyte and neutrophil proportions were elevated in the blood of all groups 2 days after infusion with $4 \times 10^4$ *L. loa* mf (3.8 ± 0.6, 9.8 ± 0.8 or 6.2 ± 3.3 mean fold increases in monocytes, 4.7 ± 0.5, 2.3 ± 0.3 or 3.0 ± 0.5-fold-increases in neutrophils, for untreated, immune-primed + untreated and immune-primed + IVM-treated mice, respectively, *n* = 5/group, all *p* < 0.05, paired *T* test, Fig. 5a). Contrastingly, the eosinophil white blood cell compartment was significantly elevated 2 days following *L. loa* infusion in IVM-treated mice only (3.1 ± 0.6 mean fold increase, *p* = 0.038, *n* = 5, paired *T* test, Fig. 5a). Tissue granulophilia was then measured within secondary lymphoid tissue or body cavities of *L. loa* microfilaraemic mice 7 days after infusion and IVM treatment, compared with untreated controls (Fig. 5b, c). In splenic tissues, a marked increase in eosinophils were apparent, associated with vasculature of the red pulp, in IVM-treated, immune-primed mice, compared with controls (2.1 ± 0.1 or 1.8 ±

0.05 mean fold change compared with naive or immune-primed sham-treated controls, respectively, $n = 5$, $p < 0.001$ one-way ANOVA with Dunnet's post-hoc test, Fig. 5b; Supplementary Fig. 6). At the same time point, within the peritoneal cavity, a significant, eosinophil-specific granulocyte exudate was manifest in immune-primed + IVM-treated mice compared with sham-treated controls $(1.0 \pm 0.15 \times 10^5$ vs. $0.32 \pm 0.06 \times 10^5$, $p < 0.05$, $n = 5$, one-way ANOVA with Dunnet's post-hoc test, Fig. 5c; Supplementary Fig. 7). These data indicate that IVM treatment responses culminate in a significant increased myelopoiesis and generalised tissue recruitment of eosinophil granulocytes in immune-primed *L. loa* microfilaraemic mice.

## Discussion

There is renewed investment in developing repurposed, reformulated or new chemical entity small molecule compounds as macrofilaricides for the priority neglected tropical disease, onchocerciasis[14,30]. A safe, short-course macrofilaricidal agent without significant direct, rapid toxicity against circulating *L. loa* mf would be a solution to the spectre of IVM-associated SAE, which is currently contributing to extended global elimination timeframes far beyond the 2025 target in countries where loasis is endemic[13]. Whilst we have made recent advances in in vitro culture screening systems for loiasis, including mf screens[31,32], the standard microfilaricidal agents: IVM, the related macrocyclic lactone, moxidectin and the piperazine derivative, diethylcarbamazine (DEC), are inactive in vitro against a range of filarial mf at concentrations known to induce efficacy in vivo[33,34] (Supplementary Fig. 4). Host-directed factors are therefore speculated to be necessary for IVM efficacy, and DEC has been demonstrated to influence host innate inflammation[33,34]. Due to the current limitations in in vivo models of loiasis, our primary goal was to establish a mouse research model that could accurately evaluate efficacies of candidate macrofilaricidal drugs on various *Loa* life cycle stages, namely developing larvae or adults in subcutaneous tissues and mf in circulation.

Our results demonstrate that development of fecund adult *Loa* infections in the natural parasitic niche can be reproducibly achieved in compound immunodeficient, lymphopenic mice lacking the common gamma chain (γc) cytokine signalling pathway. Contrastingly, following maturation, adult *Loa* survival was not dependent on γc signalling, determined by recovery of mf-producing adult parasites 1 month post implantation in γc-sufficient SCID mice. A number of cytokines (IL-2, IL-4, IL-7, IL-9, IL-15 and IL-21) signal via this shared receptor chain[35]. This suggests that facets of the murine innate immune system require one or more of the γc cytokine pathways, either during haematopoiesis or in response to infection to control loiasis throughout the L3-L4-L5 development and growth phase. Innate lymphoid cells (ILC) are known to be important innate immune cells, which require γc-dependent cytokine signalling for haematopoietic development[36]. Because specific ILC sub-sets, namely natural killer (NK) cells and ILC2, expand and are associated with immune control of specific nematode infections, including experimental filarial infections[36–39], further research is required to determine mechanistic roles for these cell types in the innate control of developing *L. loa* larvae in lymphopenic strains of mice.

The establishment of an adult loiasis research model will be of use in preclinical assessments of candidate macrofilaricides targeting *Loa* and the ability to implant defined burdens of *Loa* male and females prior to drug testing will aid accuracy of readouts. The model is thus ready to test treatment responses to 'reference' macrofilaricides, such as flubendazole, and to scrutinise whether dose alterations of human registered drugs (e.g., albendazole) or

re-purposing the veterinary agents: emodepside or oxfendazole, can mediate substantial and selective macrofilaricidal activities against *L. loa* in vivo. Because *Loa* circulating glycoproteins could be readily detected by commercial filarial immunodiagnostic tests in both pre-fecund and fecund *Loa* infected mice, this research model could also be readily applied in the preclinical discovery of *Loa* adult biomarkers and evaluation of specific *Onchocerca* candidate biomarkers currently in development as potential point-of-care diagnostics[40,41]. Beyond translational research, the murine model will be of benefit to basic parasitological researchers by facilitating a convenient, abundant source of all mammalian life cycle stages of *Loa* parasites for molecular and genomic studies as an alternative to non-human primate (NHP) usage.

The more facile approach of infusing purified mf directly into venous blood generated similar levels of parasitaemias as observed post-adult implantations, in both immunodeficient and immunocompetent mice. No model supported 'hypermicrofilariaemias' in the peripheral circulation, but displayed between $10^2$ and $10^4$ mf/ml rapidly sequestering in the cardiopulmonary circulation following $4 \times 10^4$ mf infusions. This may reflect the anatomical size differences between murine and human microvasculature, whereby *Loa* mf are hindered when traversing murine capillary beds. Alternatively, or in addition, because *Loa* exhibits a diurnal periodicity in humans[42], physiological cues for oscillatory peripheral circulatory migration versus cardiopulmonary sequestration may vary between mice and humans. Similarly, human sub-period strain *B. malayi* also demonstrate a tropism for cardiopulmonary circulation when infused into mice[21,24]. Adult splenectomy of WT mice improved parasitaemia yields yet SCID mice (without splenectomies) supported the highest parasitaemias and splenectomies of SCID mice did not further elevate microfilaraemias. This suggests that any splenic mediated clearance of mf is part of an adaptive immune-dependent process.

Consequently, we were able to validate that rapid microfilaricidal activity of IVM against blood-borne *Loa* mf could be modelled in both immunodeficient and -competent mouse strains and subsequently explored the dependence of adaptive immune processes in IVM-mediated microfilaricidal efficacy. Our data demonstrate that typical, rapid, >90% clearance of mf from both cardiopulmonary and peripheral blood by either oral or parenteral, single dose IVM does not require adaptive immunity. This was in marked contrast to a lack of in vitro activity of IVM or the related, more potent macrocyclic lactone, moxidectin, at physiologically relevant exposures of drug. However, we determined that antigen-experienced, immunocompetent mice were subtly more sensitive to the effects of IVM clearance of *Loa* mf. Our data are consistent with previous findings that IVM can mediate high microfilaricidal activities against *Brugia* or *Onchocerca* mf in lymphopenic mouse strains[21,24,43], and we conclude that whilst adaptive immunity is not a necessary host-directed component of the IVM microfilaricidal mode of action, variation in adaptive immune potential may influence magnitude of treatment response post-IVM treatment of loiasis individuals. Because we have previously developed, validated and implemented a CB.17 SCID mouse model to test macrofilaricidal activities against *Brugia* and *Onchocerca* adult filariae[21–25], the CB.17 SCID *Loa* microfilaraemic mouse model is ideal to be implemented as a corresponding counter-screen to scrutinise for 'off-target' rapid, direct microfilaricidal activities. By using the same inbred strain, significant discrepancies in drug exposures between macrofilaricide and microfilaricidal efficacy tests should be avoided. We initially implemented the screen to test for 'off-target' activities of a veterinary anthelmintic, oxfendazole, which has undergone phase I trials as a repurposed treatment for human

helminthiases (ClinicalTrials.gov NCT03035760) or the oral reformulation of flubendazole and determined that both benzimidazole anthelmintic chemotypes have no rapid IVM-like activity at dosages known to mediate macrofilaridal activity in preclinical mouse models[24,29].

Beyond the immediate translational research priorities of new drugs and diagnostics for filarial diseases in loiasis co-morbidities, the aetiology of loiasis adverse reactions needs to be more thoroughly understood. Determining the pathophysiology of Loa AE is critical if effective adjunct therapies are to be deployed during IVM administration to at-risk groups to reduce the risk of AE and increase population adherence to MDA. Adjunct therapy is foreseeable as part of a 'test-and-not-treat' strategy using the recently developed Loa CellScope point-of-care device that can discern low-moderate, high or very high microfilaraemias, as a traffic light-warning system[44,45]. We therefore exploited the susceptibility of WT mice to Loa microfilaraemias post-immune priming to initially explore the systemic inflammatory responses induced by rapid IVM-mediated clearance of mf from the blood. We chose to examine these responses in 'antigen experienced' animals to more align with chronically infected human populations. Whilst initially both type-1 (IFNγ) type-2 (IL-4, IL-5, CCL11) and regulatory type (IL-10) inflammatory mediators were upregulated post-IVM treatment, by 7 days a switch to a predominant type-2 inflammatory signature was apparent, characterised by maintenance of IL-4 and IL-5, downregulation of IFNγ and IL-10 and significant increases of the eosinophil chemotactic factor CCL11. We have further characterised an augmented eosinophilia in peripheral circulation, in secondary lymphoid tissue and in the peritoneal cavity of antigen experienced, microfilaraemic mice as a consequence of IVM treatment. This suggests a predominance of allergic immune responses are induced post IVM, probably induced by liberation of mf somatic antigens, which may be targetable via anti-allergy/asthma type therapeutics. Certainly, eosinophil-containing micro-lesions have been identified in brain capillaries post-IVM treatment of baboons[19] and both increased IL-5 and eosinophilia is evident in loiasis patients post-IVM or DEC treatment[46] corroborating that eosinophilic responses in the research model emulate clinical inflammation post treatment. Considering the availability of a full spectrum of murine biological research reagents and transgenic animals, the research model now offers a powerful new approach to dissect inflammatory AE, acute tissue pathologies and evaluate therapeutic targeting of host inflammation post-IVM treatment.

Current limitations of both research models include the proximity to a source of Loa L3, necessary continued usage of experimentally infected baboons to provide mf and restraints on the numbers of L3 that can be acquired via trapping of wild flies. This means that loiasis mouse models are not currently available outside of the endemic region of Central Africa. However, we are investigating approaches to experimentally infect Chrysops with purified mf to increase throughput of infectious stage larvae. Preliminary ongoing experiments support that microinjections of L. loa mf into wild-caught Chrysops can yield infectious stage larvae, which develop into juvenile adult worms in mice. Following ethical permissions, we intend to evaluate whether purifications of mf from hypermicrofilaraemic human donors can be used to obviate the requirements for NHP experimentation and further increase throughput. Such experiments may also define the extent of diurnal periodic migrations displayed by blood-borne mf within microfilaraemic mouse models. Certainly, preliminary studies in splenectomised baboons indicate diurnal fluctuations of human strain L. loa are apparent. Further, we are validating whether using cryopreservations of mf or L3 can extend the accessibility of the models to non-endemic country research laboratories. Another present limitation of the infusion

model to study treatment-associated pathology is that overt neurological-type AE has not yet been observed post-IVM delivery. However, we are currently investigating the fate of mf post treatment, including tropism and histopathological consequences in brain-associated vasculature, and whether increasing inoculates of mf in SCID or WT mice will lead to evidence of neurological dysfunction post treatment.

For basic biology research of loiasis and other filariae, establishing mouse models will be a powerful new tool to interrogate important host–parasitological interactions, such as how periodicity is influenced by host factors and dissecting stage-specific immune responses to larvae, adults and mf. Similarly, by establishing microinjections of mf derived from mice into Chrysops vectors, detailed, timed infection-courses could reveal new biological insights of the parasite–vector biology.

In conclusion, we have developed novel small animal models of loiasis sufficiently robust for immediate implementation in preclinical research to accelerate development of novel drugs, therapeutics and diagnostics for filariasis elimination in Central Africa.

## Methods

**Animals.** BALB/c, BALB/c RAG2$^{-/-}$γc$^{-/-}$, CB.17 SCID, NOD.SCID and NOD.SCIDγc$^{-/-}$ (NSG) mice of 5–6 weeks of age were purchased from Charles River Europe, and BALB/c RAG2$^{-/-}$ mice were kindly provided by Prof Andrew McKenzie (MRC Laboratory of Molecular Biology, Cambridge University, UK) and by Prof. Dr. Antonius Rolink (Developmental and Molecular Immunology Department of Biomedicine, University of Basel, Switzerland). Mice were shipped in filter topped boxes to Research Foundation for Tropical Diseases and the Environment (REFOTDE), Buea, Cameroon. All mice were reared in REFOTDE in a 12:12 light:dark cycle, maintained in individually ventilated caging (IVC) with HEPA filtered air system (Tecniplast) with ad libitum provision of standard irradiated rodent chow and bottled mineral water. All mice used for the experiments were infected in the laboratory of REFOTDE. All animal procedures received ethical approval by the Animal Care Committee at REFOTDE and undertaken in accordance with UK regulatory standards.

L. loa microfilariae (mf) were obtained from splenectomised infected baboons (Papio anubis) that were kept in captivity and infected with the L. loa human strain[17,32,47]. The acquisition, care and ethical concerns on the use of baboons as donors of mf have been previously documented[18,20]. Ethical and administrative approvals for the use of baboons in this study were obtained from the Ministry of Scientific Research and Innovation of Cameroon (Research permit #028/MINRESI/B00/C00/C10/C12) and the Animal Care Committee at REFOTDE. Procedures adhered to the NIH Guide for the Care and Use of Laboratory Animals. The authors have complied with all relevant ethical regulations for animal testing and research.

**Parasites.** L. loa mf were extracted from the blood using a Percoll® gradient centrifugation processing[18,32,48] with a discontinuous gradient of 65%, 50% and 40% iso-osmotic Percoll®. Two milliliter of whole blood collected on the same day were loaded on the top of the gradient and centrifuged at 400 g for 20 min; the layer containing mf was carefully removed and filtered gently through a 5- μm pore-size cellulose filter, and mf were then transferred into a Petri dish containing the culture medium (10% foetal calf serum (FCS) supplemented Dulbecco's modified Eagle's medium (DMEM) and incubated 5 min/37 °C to release mf. Suspensions were then centrifuged to concentrate mf. Unless stated otherwise, $4 \times 10^4$ mf in 150 μl of RPMI were loaded in 26 G 1 -ml insulin syringes and maintained at 37 °C until intravenously injected in the tail vein of the mice.

L. loa L3 infective stages were derived from wild-caught Chrysops silacea via baited traps in a known hyperendemic area. Flies were dissected to allow release of any L. loa L3 infective stages. Infective doses of 100–200 L3 per 200 μL of medium (DMEM + 10% FCS) were loaded in 25 G 1 -mL syringes and subcutaneously injected into mice.

L. Loa adult worms were obtained from CB.17 SCID, NOD.SCID, NOD.SCIDγc$^{-/-}$ (NSG) and BALB/c RAG2$^{-/-}$γc$^{-/-}$ mice that were infected with L. loa L3 and culled either at 3 or 5 months post infection. Worms were then either subcutaneously re-implanted into new recipient mice, fixed for histology or utilised in embryogram assays.

**Experimental infections and drug treatments.** Mice were infected with either 100 or 200 L.loa L3 via subcutaneous injection or with 5 male/5 female L. loa adult worms via surgical subcutaneous implantation or $4 \times 10^4$ L. loa mf via a tail vein infusion. In the case of surgical implantation, mice were placed under anaesthesia using intra-peritoneal injections of ketamine (Ketaset, 70 mg/kg) and medetomidine (Domitor, 0.8 mg/kg) with surgery, post-operative monitoring and recovery[21]. L. loa adults were washed in several changes of the medium (DMEM + 10% FCS),

and groups of five females and five males were surgically implanted under the dorsal back skin behind the neck.

For immune priming experiments, aliquots of $3 \times 10^4$ *L. loa* mf were killed through three cycles of 10 min at $+50\,°C$ followed by 10 min at $-50\,°C$ incubations. Aliquots of $3 \times 10^4$ dead *L. loa* mf were subcutaneously injected to each mouse. Mice were challenged with $4 \times 10^4$ *L. loa* mf 14 days post-immune priming.

Mice were treated orally with ivermectin (Sigma-Aldrich) at 1 mg/kg in water, with ivermectin subcutaneous injections at 5 mg/kg in water, orally with flubendazole (Janssen BEND formulation, a gift from Dr B. Baeton, Jansssen Pharmaceutica, Beerse, Belgium) at 40 mg/kg[24,29], or orally with oxfendazole (Dolthene formulation, Vetsend) at 25 mg/kg in corn oil.

Mice were euthanized using $CO_2$ rising concentrations. Post-mortem, blood was collected by cardiac puncture using either a heparinized 25 G needle and 1-mL syringe for plasma collection or a non-heparinized syringe and needle for thick smears and serum collection followed by systematic dissection for adult parasite recovery.

**Peripheral and cardiopulmonary microfilaraemias.** Infected mice were checked for the presence of mf in blood by thick smear and subsequent Giemsa staining. Briefly, blood was collected from the tail vein ($2 \times 20\,\mu L$) or from the heart ($2 \times 50\,\mu L$) at necropsy using a 25 G 1-mL syringe, transferred onto glass slides and then processed for thick smears through a scratch method. Air-dried smears were incubated in distilled water for 4 min to lyse erythrocytes, fixed in methanol for 1 min and finally stained with 40% Giemsa for 40 min then washed in distilled water until clear. Duplicate slides were counted twice with microscope operators blinded to treatment group.

**Serological studies.** Mouse sera or plasma were tested for filarial antigen detection using two kits commercially available for the detection of *Wuchereria bancrofti* antigens: the filariasis test strip (FTS, Alere, Abbott, UK) and the Og4C3 filariasis antigen ELISA (Tropbio, Cellabs, UK), both as per the manufacturer's instructions. A volume of 70 µLof fresh serum was used immunochromatographic detection with the FTS per test strip with reading taken within minutes after sample application.

Plasma cytokines/chemokines levels in mouse plasma were determined using a 32-analyte multiplex cytokine immunoassay based on xMAP technology (MCYTMAG-70K-PX32 kit, Millipore) as per the manufacturer's instructions and samples were analysed on a LX100$^{TM}$ multiplexing instrument (Luminex). Analytes included were: eotaxin, G-CSF, GM-CSF, M-CSF, gamma interferon (IFN-γ), TNF-α, IL-1α, IL-1β, IL-2, IL-3, IL-4, IL-5, IL-6, IL-7, IL-9, IL-10, IL-12 (p40), IL-12 (p70), IL-13, IL-15, IL-17, IP-10, CCL2, CCL3, CCL4, CCL5, CXCL1, CXCL2, CXCL5, CXCL9, leukaemia inhibitory factor (LIF) and vascular endothelial growth factor (VEGF).

**Flow cytometry.** Mouse peritoneal cells were collected via a peritoneal cavity wash with 10 mL of PBS-5% FCS. Single-cell suspensions were prepared in FACS buffer (PBS + 0.5% BSA + 2 mM EDTA). Fc receptors were blocked with αCD16/32 (1/40 dilution, ref: 14-0161-82, eBioscience) prior to the application of the following cocktail: viability dye eFluor450 (ref: 65-0863-14, eBioscience), anti-mouse SiglecF-APC (dilution 1/40, clone S17007L, ref: 155508, Biolegend) and anti-mouse Ly6G-FITC (dilution 1/40, clone RB6-8C5, ref: 11-5931-82, eBioscience) or their matched isotype controls using a fluorescence-minus-one method. Samples were fixed in FACS buffer containing 0.5% PFA and shipped at 4 °C to UK. All multi-labelled cell samples were subsequently acquired using a BD LSR II flow cytometer (BD Bioscience) and analysed on FlowJo Software (Supplementary Fig. 7). OneComp eBeads (eBioscience) were used to optimise antibody staining panels and apply compensation. For compensation controls, optimal PMT voltages for the positive signal to be detected were set within $10^4$ and $10^5$ whereas negative signal was set to below $10^2$.

**Differential blood count thin smears.** Thin smears were performed on peripheral blood collected from the mouse tail vein and were stained in May–Grünwald (Sigma) for 5 min then in 40% Giemsa for 30 min. Leucocytes numeration was performed under a microscope counting 200 leucocytes. Operators were blinded to mouse groups.

**Histological studies.** Female and male worms were processed as whole mounts for anatomical observation. Worms were fixed in hot 70% ethanol then mounted on a glass slide within drops of glycerol. Samples were allowed an overnight incubation a room temperature to fully perfuse the worms with glycerol, and worms were analysed on an Olympus BX60 microscope.

Spleens from mice were collected at readout and fixed in 10% PFA for 24 h then transferred in 70% ethanol and subsequently embedded in paraffin. Paraffin sections (5 µm) were stained with hematoxylin-eosin (H&E, VWR) for the detection of eosinophils. Images were acquired using the digital slide scanner HPF-NanoZoomer RS2.0 (Hamamatsu) coupled to a high definition 3-CCD digital camera of the PHIC immunohistochemistry platform (Institut Paris-Saclay d'Innovation Thérapeutique, France). Eosinophils were counted from 10 quadrants of 120 µm x 200 µm per splenic zone per mouse. Operator was blinded to mouse group.

**Statistical analysis.** Data were first checked for normality, using D'Agostino & Pearson omnibus Shapiro–Wilk normality tests before or post log$_{10}$ transformation of raw data. If raw or transformed data passed both tests, two-tailed parametric Student's *T*-Test (two variables) or one-way ANOVA with Dunnett's post-hoc tests (> two variables) were used. If the data failed to pass normality testing, two-tailed Mann–Whitney (two groups) or Kruskal–Wallis with Dunn's post-hoc tests (>two variables) were used. Significant changes in paired data were tested by *T* tests or Wilcoxon tests. All tests were performed in GraphPad Prism software at a significance level of 5% and significance is indicated: $*p < 0.05$, $**p < 0.01$, $***p < 0.001$. Heatmap analysis was undertaken in Excel using conditional formatting with $-10$-fold change being colour-coded as blue and $+10$ fold change being reported as red.

**Reporting summary.** Further information on experimental design is available in the Nature Research Reporting Summary linked to this article.

## Data availability

The data that support the findings of this study are available from the corresponding author upon reasonable request.

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

## Acknowledgements
The authors would like to thank Benny Baeton and Marc Englen (Janssen Pharmaceutica, Beerse) for the donation of oral flubendazole formulation, Dr Marc Hubner (University Clinic Bonn) for helpful advice on oxfendazole oral dosing, Prof. Andrew McKenzie (MRC Laboratory of Molecular Biology, Cambridge University, United Kingdom) and Prof. Dr. Antonius Rolink (Developmental and Molecular Immunology Department of Biomedicine, University of Basel, Switzerland) for providing BALB/c RAG2⁻/⁻ mice and Oben Bruno for husbandry of experimental baboons. We also thank Francoise Gaudin and the PHIC immunohistochemistry platform (Institut Paris-Saclay d'Innovation Thérapeutique, France). This work was supported by a Bill and Melinda Gates Foundation Grand Challenges Explorations Grant (OPP1119043) to J.D.T., S.W. and M.J.T.

## Author contributions
Conceptualisation: J.D.T., S.W. and M.J.T. Methodology: J.D.T., S.W., N.P.P. and H.S. Formal analysis: J.D.T., S.W., N.P.P., H.S., H.M.M. and V.C.C. Resources: J.D.T. and S.W. Writing: N.P., J.D.T., H.S. and S.W. Investigation: N.P., H.S., H.M.M., V.C.C., A.J.N., F.F.F., D.B.T., N.V.G., D.N.A., P.W.C., B.L.N., S.W. and J.D.T. Funding acquisition: J.D.T., S.W. and M.J.T.

## Additional information

**Competing interests:** The authors declare no competing interests.

