## [Peer Review File · Nature Communications]

Reviewers' Comments:

Reviewer #1:

Remarks to the Author:

The mouse models described in this manuscript provide significant value to the community of scientists working to eliminate the major human filariases. The experiments were well-designed and analyzed and the authors have drawn reasonable conclusions from the results. I have only minor suggestions for improvement.

1. Introduction, 10th line from the top: I believe the sentence should read, "...adverse events SAEs) may occur.....", as they are not inevitable in this set of patients.
2. Page 3, 4th line: I believe it should be "mandrill" rather than "drill"
3. Results section, last paragraph: Please delete "marginally non-significantly increased" and "non-significantly decreased" and the related results. The correct interpretation is that these factors did not change under these experimental conditions.

Reviewer #2:

Remarks to the Author:

This is an excellent paper describing the development of mouse model for *Loa loa*. The studies are well designed and extensive. Observations include description of a fully permissive model for *L. loa*, treatment of the infection with anthelmintics and analysis of inflammatory responses. The work is novel and will have a significant impact on the field.

Minor comments:

1. It would be helpful if the authors would clarify the mouse strains used in this study and compare them to the commonly used NSG, NOG and NRG mice.
2. Were eosinophils seen as part of the inflammatory response in the infected mice? This would be an important corollary to the clinical presentation.
3. Was there periodicity seen in the microfilaraemias? This would also help define the new model.

Reviewer #3:

Remarks to the Author:

General Comments

This manuscript demonstrates, by using immunodeficient mice lacking the common gamma-chain, that a mouse model that is permissive for the entire life cycle of *Loa loa* can be established. Using SCID or RAG gamma-chain deficient mice infected experimentally with infective larvae of *Loa loa*, not only can adult female and male worms develop, but they can mate and produce microfilariae (mf). At the same time, the authors show that in immunocompetent mice, iv infusion of mf can be used, in the short term as a model for *Loa loa* microfilaraemia, allowing for studies related to drug treatment and host responses in mf positive mice. This latter approach extends to *Loa loa* mf what has been done previously by some of the authors in related filarial infections.

While the studies reported are both interesting and important, there is not a single biologic theme, but rather only a unifying organism (*Loa loa*). Moreover, the data reported are a bit of a hodgepodge of disparate findings leaving the reader without a clear message apart from the ability to complete the *Loa loa* cycle in the NOD/SCID/γc or NOD/RAG/γc mice. It is not clear whether the message is about the host response (not really examined), drug treatment, adverse events following ivermectin, cross reactivity with assays of circulating filarial antigen or something else.

Specific issues/concerns

1) Related to the establishment of the complete model

- a. It would have been nice to demonstrate that the Loa could be transmitted to naïve Chrysops following the development of microfilaraemia
- b. Were there any perturbations of circulating myeloid cells in the Loa-infected immunodeficient mice? In particular, did infection drive eosinophilia as is commonly seen in these infections?
- c. Did the mf of Loa in the permissive model retain their diurnal periodicity? This would be quite important to understand whether the cues for the periodicity are host or parasite-derived.
- d. Although the authors argue that this model is a good one for the study of drug effects, why not treat these infected-mice with drugs known to have some macrofilaricidal activity (e.g. albendazole or even diethylcarbamazine) and/or microfilaricidal activity
- e. The authors need to put this study in the context of the other helminth infections that have used these mouse strains to make permissive an infection otherwise less-permissive (for example *Strongyloides stercoralis*). In addition, the role of NK cells that are likely the problem when comparing the γ c knockout strains with those with γ c intact should be addressed, particularly in the context of the Rajan et al studies with *Brugia*.

2) Related to the studies of microfilaraemia

- a. How is this very different from the studies performed in the 1990s by one of the co-authors using *Brugia* and *Onchocerca lienalis*?
- b. In the data shown following ivermectin in this model (Figure 4), are there any data on the consequences of this post-treatment Type -2 inflammatory response. Were there post-treatment changes in any of the peripheral leukocytes? Was there any histology done on the lymph nodes (or cardiac muscle) following ivermectin administration? This might allow for insights into the induction of serious adverse events in *Loa loa* following microfilaricidal drug administration.

Reviewers' comments:

Reviewer #1 (Remarks to the Author):

The mouse models described in this manuscript provide significant value to the community of scientists working to eliminate the major human filariases. The experiments were well-designed and analyzed and the authors have drawn reasonable conclusions from the results.

We thank the reviewer for this very positive feedback.

I have only minor suggestions for improvement.

1. Introduction, 10th line from the top: I believe the sentence should read, "...adverse events (SAEs) may occur.....", as they are not inevitable in this set of patients.

This has been amended accordingly.

2. Page 3, 4th line: I believe it should be "mandrill" rather than "drill"

The species, *Mandrillus leucophaeus*, is commonly referred to as 'the drill', which is related to *Mandrillus sphinx*, 'the mandrill'.

3. Results section, last paragraph: Please delete "marginally non-significantly increased" and "non-significantly decreased" and the related results. The correct interpretation is that these factors did not change under these experimental conditions.

We have deleted text referring to marginally non-significant data ($\alpha < 0.1 > 0.05$). Because these analytes were up or down-regulated on average by ≥ 2 -fold, we retain the associated analyses.

Reviewer #2 (Remarks to the Author):

This is an excellent paper describing the development of mouse model for *Loa loa*. The studies are well designed and extensive. Observations include description of a fully permissive model for *L. loa*, treatment of the infection with anthelmintics and analysis of inflammatory responses. The work is novel and will have a significant impact on the field.

We thank the reviewer for recognizing the importance and novelty of the work

Minor comments:

1. It would be helpful if the authors would clarify the mouse strains used in this study and compare them to the commonly used NSG, NOG and NRG mice.

Full details of strains used and commercial supply can be found in the materials and methods.

We have clarified that NOD.SCID γ ^{-/-} is more commonly referred to as NSG in the methods and the results.

2. Were eosinophils seen as part of the inflammatory response in the infected mice? This would be an important corollary to the clinical presentation.

Following reviewer 2 and reviewer 3 comments, an additional investigation has been conducted examining eosinophilia in peripheral blood, secondary lymphoid tissue and peritoneal cavity. A new figure has been created, reporting proportional increases in eosinophils in peripheral blood as well as significant tissue eosinophilia in splenic secondary lymphoid tissues and serous cavities of immune-primed+IVM treated mice (Figure 5). We have summarized this additional data in a final paragraph in the results section:

“Differential white blood cell counts were undertaken at baseline and D2. Both monocyte and neutrophil proportions were elevated in the blood of all groups two days after infusion with 4×10^4 L. loa mf (3.8±0.6, 9.8±0.8 or 6.2±3.3 mean fold-increases in monocytes, 4.7±0.5, 2.3±0.3 or 3.0±0.5 fold-increases in neutrophils, for untreated, immune-primed+untreated and immune-primed+IVM-treated mice, respectively, n=5 / group, all p<0.05, Fig. 5A). Contrastingly, the eosinophil white blood cell compartment was significantly elevated 2 days following L. loa infusion in IVM-treated mice only (3.1±0.6 mean fold-increase, p=0.038, Fig 5A). Tissue granulophilia was then measured within secondary lymphoid tissue or body cavities of L. loa microfilaraemic mice seven days after infusion and IVM treatment, compared with untreated controls (Fig. 5B&C). In splenic tissues, a marked increase in eosinophils were apparent, associated with vasculature of the red pulp, in IVM-treated mice, compared with controls (2.1±0.1 or 1.8±0.05 mean fold-change compared with Fig. 5B and supplementary figure 6) At the same time-point, within the peritoneal cavity, a significant, eosinophil-specific granulocyte exudate was manifest in IVM-treated mice compared with sham treated controls ($1.0 \pm 0.15 \times 10^5$ versus $0.32 \pm 0.06 \times 10^5$ p<0.05, Fig. 5C and supplementary figure 7). These data indicated IVM treatment responses culminated in a significant increased myelopoiesis and general tissue recruitment of eosinophil granulocytes in immune-primed L. loa microfilaraemic mice.”

3. Was there periodicity seen in the microfilaremias? This would also help define the new model.

We thank the reviewer for this very interesting question. We agree that additional investigations on periodicity could be an onward use of the research model if validated. The mf used in the experiments presented are sourced from long-term baboon experimental infections with human strain *L. loa*. To be accurate about periodic parasitological migrations in the mouse model, more extensive investigations would be required with ethical permissions to acquire diurnal periodic mf from infected human volunteers. This is beyond the scope of the current paper.

Reviewer #3 (Remarks to the Author):

General Comments

This manuscript demonstrates, by using immunodeficient mice lacking the common gamma-chain, that a mouse model that is permissive for the entire life cycle of *Loa loa* can be established. Using SCID or RAG gamma-chain deficient mice infected experimentally with infective larvae of *Loa loa*, not only can adult female and male worms develop, but they can mate and produce microfilariae (mf). At the same time, the authors show that in immunocompetent mice, iv infusion of mf can be used, in the short term as a model for *Loa loa* microfilaraemia, allowing for studies related to drug treatment and host responses in mf positive mice. This latter approach extends to *Loa loa* mf what has been done previously by some of the authors in related filarial infections.

While the studies reported are both interesting and important, there is not a single biologic theme, but rather only a unifying organism (*Loa loa*). Moreover, the data reported are a bit of a hodgepodge of disparate findings leaving the reader without a clear message apart from the ability to complete the *Loa loa* cycle in the NOD/SCID/ γ c or NOD/RAG/ γ c mice. It is not

clear whether the message is about the host response (not really examined), drug treatment, adverse events following ivermectin, cross reactivity with assays of circulating filarial antigen or something else.

We thank the reviewer for their considered comments. Because this is a report of an entirely new preclinical research model to address several important issues hindering filariasis elimination efforts, we intentionally undertook a broad series of experiments to test the translational medicine utility of the model. Therefore, the ‘unifying theme’ of the paper is to consider each problem area in turn (namely: affordable and scalable point-of-care diagnostics discriminating specific filarial infections, treatment responses to new candidate anti-filarial drugs, and mechanisms of *Loa* hypermicrofilaraemia treatment adverse events). In each area, we provide sufficiently robust evidence that the new research models in mice are innovative, advantageous, novel tools which can now be provided to academic and industry research groups to tackle this urgent global health problem.

Specific issues/concerns

1) Related to the establishment of the complete model

a. It would have been nice to demonstrate that the *Loa* could be transmitted to naïve *Chrysops* following the development of microfilaraemia

We agree this would be an important next step. A companion paper is being readied for submission which illustrates success in artificial infection (micro-injection) of wild caught *Chrysops* to this end.

b. Were there any perturbations of circulating myeloid cells in the *Loa*-infected immunodeficient mice? In particular, did infection drive eosinophilia as is commonly seen in these infections?

We now provide extra data from a single experiment at the request of reviewer 2 and 3 detailing significant alterations in the myeloid cell compartment including a significant eosinophilia post-IVM treatment (Figure 5).

c. Did the mf of *Loa* in the permissive model retain their diurnal periodicity? This would be quite important to understand whether the cues for the periodicity are host or parasite-derived.

Please see response to reviewer 2

d. Although the authors argue that this model is a good one for the study of drug effects, why not treat these infected-mice with drugs known to have some macrofilaricidal activity (e.g. albendazole or even diethylcarbamazine) and/or microfilaricidal activity

We thank the reviewer for their insight. We agree that the adult *Loa* implant model is certainly ready to screen macrofilaricidal dose regimens known to be efficacious against other medically-important filariae in preclinical testing. This would require major investment and time (6 months from inception to readout), most likely to be supported from industrial sponsorship. We argue that publishing our model development as soon as possible in a high impact journal will maximize publicity and advocacy for onward adoption to screen new treatments for loiasis. We extend our discussion to advocate use of the model in this regard:

“The establishment of an adult loiasis research model will be of use in preclinical assessments of candidate macrofilaricides targeting Loa and the ability to implant defined burdens of Loa male and females prior to drug testing will aid accuracy of readouts. The model is thus ready to test treatment responses to ‘reference’ macrofilaricides, such as flubendazole, and to scrutinise whether dose alterations of human registered drugs (e.g. albendazole) or re-purposing veterinary agents (emodepside, oxfendazole) can mediate substantial and selective macrofilaricidal activities against L. loa in vivo.” Discussion, third paragraph

e. The authors need to put this study in the context of the other helminth infections that have used these mouse strains to make permissive an infection otherwise less-permissive (for example *Strongyloides stercoralis*). In addition, the role of NK cells that are likely the problem when comparing the γ c knockout strains with those with γ c intact should be addressed, particularly in the context of the Rajan et al studies with *Brugia*.

We thank the reviewer for those valuable comments. We indeed are assessing the role of innate lymphoid cell subsets including NK cells in innate immune control of medically-important filariae in mice. Our manuscript therefore lays the foundations for further work to be conducted in the matter, specifically regarding loiasis infection.

Whilst we originally alluded to γ c processes in innate lymphoid cell haematopoiesis, we now add to the discussion to more specifically propose that ILCs, including but not limited to NK cells, may be an important residual innate cell type able to control filarial larval establishment and immature adult development in mice.

“Innate lymphoid cells (ILC) are known to be important innate immune cells which require γ c-dependent cytokine signalling for haematopoietic development³⁶. Because specific ILC sub-sets, namely Natural Killer (NK) cells and ILC2, expand and are associated with immune control of specific nematode infections, including experimental filarial infections³⁶⁻³⁹, further research is required to determine mechanistic roles for these cell types in the innate control of developing immature L. loa in lymphopenic strains of mice.” Discussion, second paragraph

2) Related to the studies of microfilaraemia

a. How is this very different from the studies performed in the 1990s by one of the co-authors using *Brugia* and *Onchocerca lienalis*?

The infusion models reported here are *Loa* specific which are essential to accurately determine any impact of novel candidate filaricides in inducing rapid ‘ivermectin-like’ dynamics of clearance of *Loa* mf from circulation. Clearly, differences may be manifest, in example, anti-*Wolbachia* drugs where the *Loa* parasite lacks the symbiosis and therefore any related drug target. For regulatory approval of new indications (especially those targeting nematode specific molecules) empirical *in vivo* safety evaluations against *Loa* mf are necessary to provide a degree of confidence they are safe in *Loa* hypermicrofilaraemic individuals.

Further, considering adverse inflammatory reactions, a *Loa* specific research model is required to more accurately emulate the host-response to *Loa* specific antigens (and without *Wolbachia* confounding responses) released by disintegrating mf.

b. In the data shown following ivermectin in this model (Figure 4), are there any data on the

consequences of this post-treatment Type -2 inflammatory response. Were there post-treatment changes in any of the peripheral leukocytes? Was there any histology done on the lymph nodes (or cardiac muscle) following ivermectin administration? This might allow for insights into the induction of serious adverse events in Loa loa following microfilaricidal drug administration.

We thank the reviewer for this comment; further investigation has been conducted on whether multiplex data would be corroborated by any systemic marked eosinophilia or in lymphoid tissues (please also see comments to reviewer 2). As indicated in Figure 5, a peak in circulating neutrophils and monocytes was detected in peripheral blood of all animal groups 2 days following infusion. However, significant increases in eosinophils were only detected post-IVM treatments. In addition, we examined splenic tissues via histology and recorded elevated eosinophils associated with splenic vessels within the red pulp at day 7 post-IVM treatment compared with non-treated and infused groups. Further we were able to perform flow cytometry on peritoneal washings and identified increases in eosinophils but not neutrophils in IVM treated animals.

We now provide this extra data as an additional figure in the manuscript (figure 5) and two additional supplementary figures (5&6).

Reviewers' Comments:

Reviewer #2:

Remarks to the Author:

The authors have responded appropriately to all of the reviewers comments

Reviewer #3:

Remarks to the Author:

The authors have largely addressed the reviewers' comments adequately and in in one instance provided information. There are several items that could have been answered more clearly (and with data).

- 1) First was the issue of infectivity of the parasites for the Chrysops vector (preparing a second publication). At least a sentence in the Discussion about the utility of this model for study host-vector interactions should be made with a reference to unpublished data.
- 2) The issue of periodicity remains an important biological question for all of the filariae. It should be simple (given the model's utility) to assess the periodicity of the microfilariae. At the least, the authors should be able to allude to the primate information from which the microfilariae (and then the infective stage parasites) were derived.

Reviewer #3 (Remarks to the Author):

The authors have largely addressed the reviewers' comments adequately and in one instance provided information. There are several items that could have been answered more clearly (and with data).

1) First was the issue of infectivity of the parasites for the *Chrysops* vector (preparing a second publication). At least a sentence in the Discussion about the utility of this model for study host-vector interactions should be made with a reference to unpublished data.

2) The issue of periodicity remains an important biological question for all of the filariae. It should be simple (given the model's utility) to assess the periodicity of the microfilariae. At the least, the authors should be able to allude to the primate information from which the microfilariae (and then the infective stage parasites) were derived.

To address the reviewer's requests, we have inserted the highlighted text into the third from final and penultimate paragraphs of the discussion:

*Current limitations of both research models include the proximity to a source of *Loa* L3, necessary continued usage of experimentally infected baboons to provide mf and restraints on the numbers of L3 that can be acquired via trapping of wild flies. This means that loiasis mouse models are not currently available outside of the endemic region of Central Africa. However, we are investigating approaches to experimentally infect *Chrysops* with purified mf to increase throughput of infectious stage larvae. Our unpublished experiments thus far support that micro-injections of *L. loa* mf into wild-caught *Chrysops* can yield infectious stage larvae which develop into juvenile adult worms in mice. Following ethical permissions, we intend to evaluate whether purifications of mf from hypermicrofilaraemic human donors can be used to obviate the requirements for NHP experimentation and further increase throughput. Such experiments may also define the extent of diurnal periodic migrations displayed by blood borne mf within microfilaraemic mouse models. Certainly, preliminary*

studies in splenectomised baboons indicates diurnal fluctuations of human strain L .loa are apparent. Further, we are validating whether using cryopreservations of mf or L3 can extend the accessibility of the models to non-endemic country research laboratories. Another present limitation of the infusion model to study treatment-associated pathology is that overt neurological-type AE have not yet been observed post-IVM delivery. However, we are currently investigating the fate of mf post-treatment, including tropism and histopathological consequences in brain-associated vasculature, and whether increasing inoculates of mf in SCID or WT mice will lead to evidence of neurological dysfunction post-treatment.

For basic biology research of loiasis and other filariae, establishing mouse models will be a powerful new tool to interrogate important host-parasitological interactions such as how periodicity is influenced by host factors and dissecting stage-specific immune responses to larvae, adults and mf. Similarly, by establishing microinjections of mf derived from mice into Chrysops vectors (our unpublished data), detailed, timed infection-courses could reveal new biological insights of the parasite-vector biology.